# Pinpointing Functionally Relevant miRNAs in Classical Hodgkin Lymphoma Pathogenesis

**DOI:** 10.3390/cancers16061126

**Published:** 2024-03-12

**Authors:** Yujia Pan, Roza Cengiz, Joost Kluiver, Arjan Diepstra, Anke Van den Berg

**Affiliations:** Department of Pathology and Medical Biology, University of Groningen, University Medical Center Groningen, 9713 RB Groningen, The Netherlands; y.pan@umcg.nl (Y.P.); r.cengiz@umcg.nl (R.C.); j.l.kluiver@umcg.nl (J.K.); a.diepstra@umcg.nl (A.D.)

**Keywords:** classical Hodgkin lymphoma, miRNA, loss of B-cell phenotype, immune evasion, growth support

## Abstract

**Simple Summary:**

We explored how small non-coding RNAs, known as miRNAs, are involved in the pathogenesis of cHL, a lymphoma originating from B cells. While miRNAs have shown broad effects in normal cellular processes and cancers, their contribution to cHL has been less explored. We organized the published human miRNA profiling studies of cHL and selected genes that are crucial for cHL pathogenesis such as those leading to the loss of B-cell phenotypes, immune evasion, and promotion of tumor growth. By providing a detailed analysis of the interactions between these miRNAs and their target genes, our review not only enhances the understanding of cHL molecular mechanisms but also paves the way for further research into how specific miRNAs could be involved in cHL progression.

**Abstract:**

Classical Hodgkin lymphoma (cHL) is a hematological malignancy of B-cell origin. The tumor cells in cHL are referred to as Hodgkin and Reed–Sternberg (HRS) cells. This review provides an overview of the currently known miRNA–target gene interactions. In addition, we pinpointed other potential regulatory roles of microRNAs (miRNAs) by focusing on genes related to processes relevant for cHL pathogenesis, i.e., loss of B-cell phenotypes, immune evasion, and growth support. A cHL-specific miRNA signature was generated based on the available profiling studies. The interactions relevant for cHL were extracted by comprehensively reviewing the existing studies on validated miRNA–target gene interactions. The miRNAs with potential critical roles included miR-155-5p, miR-148a-3p, miR-181a-5p, miR-200, miR-23a-3p, miR-125a/b, miR-130a-3p, miR-138, and miR-143-3p, which target, amongst others, PU.1, ETS1, HLA-I, PD-L1, and NF-κB component genes. Overall, we provide a comprehensive perspective on the relevant miRNA–target gene interactions which can also serve as a foundation for future functional studies into the specific roles of the selected miRNAs in cHL pathogenesis.

## 1. Introduction

Hodgkin lymphoma (HL) is a hematological malignancy characterized by a small number of tumor cells in the affected tissues. The tumor cells originate from germinal center (GC) B cells that have escaped from apoptosis during the selection process in the germinal centers [1]. The World Health Organization (WHO) classification includes two main subtypes of HL, i.e., classical HL (cHL), accounting for 95% of all cases, and nodular lymphocyte-predominant HL (NLPHL) [2].

In about 30% of cHL cases, the tumor cells—referred to as Hodgkin and Reed–Sternberg (HRS) cells—are infected by the Epstein–Barr virus (EBV). In these cases, EBV-derived proteins are considered to drive transformation [3]. Regardless of EBV status, HRS cells have lost most of their B-cell phenotypes, exemplified by loss of markers such as CD19, CD20, and CD79A/B [4,5]. This has been attributed to the downregulation or loss of expression of several transcription factors (TFs), such as PAX5, OCT2, PU.1, EBF1, and ETS1 [6,7]. Moreover, HRS cells overexpress CD30 which is usually expressed only by small subpopulations of B cells at distinct differentiation stages [8,9]. CD30 has been recognized as a diagnostic marker and therapeutic target in cHL [10].

HRS cells are surrounded by an abundant infiltrate of immune cells that constitute an extensive tumor microenvironment (TME). The TME is ineffective in inducing anti-tumor cell responses because HRS cells employ diverse immune evasive mechanisms. One commonly observed mechanism is the loss of HLA class I and class II which are essential for presenting antigens to immune cells and initiating an immune response [11,12]. In addition, HRS cells can express programmed death ligand 1 (PD-L1) and PD-L2 which both engage in PD-1 signaling and induce T-cell “exhaustion” [13]. The PD-L1–PD-1 axis is a well-known therapeutic target for treating cHL patients [14]. About half of cHL cases also express PD-L2, which also engages in tumor cell immune evasion [14].

Besides escape from immune cells, HRS cells rely on aberrant activation of pathways such as NF-kB, JAK/STAT, Notch, PI3K/AKT, and MAPK for survival and proliferation [15]. For example, high expression of CD40 on HRS cells is crucial for the activation of NF-κB, which is essential for tumor cell survival through the induction of anti-apoptotic genes [10,16]. Moreover, multiple cytokines and chemokines like IL-3, IL-7, IL-9, IL-13, CCL17, and CCL22 provide direct growth support, attract growth-supporting immune cells, and help to block activity of cytotoxic immune cells [16].

Although much insight has been gained regarding the function of protein-coding genes, the current knowledge on the role of miRNAs targeting these proteins remains limited. miRNAs inhibit the translation of their target mRNA molecules by base-pairing to complementary binding sites usually present in the 3′ untranslated region (3′UTR) [17]. Often, the inhibition of translation by miRNAs is accompanied by degradation of the target mRNA. Base-pairing of nucleotides 2–7 of the miRNA seed region to the target mRNA is critical for effective inhibition [18]. miRNAs have been shown to be crucially involved in almost all biological processes [19,20].

The first miRNA identified in cHL was the non-coding RNA known as BIC. This primary miRNA gene (miR155HG) encoding miR-155 was shown to be highly expressed in cHL cell lines and primary HRS cells [21,22]. Since then, multiple studies have reported on the miRNA landscape in cHL, including EBV-derived miRNAs [23,24]. Although several studies reported miRNAs that are abundantly or differentially expressed in HRS cells, their functional relevance for the pathogenesis of cHL remains largely unknown. For a limited number of miRNAs, target genes have been identified in cHL cell lines. Most striking are those for miR-155 in cHL cell lines, including AGTR1, ZNF537, FGF7, ZIC3, MAF, DET1, NIAM, HOMEZ, PSIP1, and JARID2 [25,26]. miR-9 and let-7a were shown to target PRDM1 (BLIMP1), CD99, DICER1, and HuR (ELAVL1) [27,28,29]. The targets of the miR-17/106b family include YES1, RBJ, NPAT, FBX031, OBFC2A, GPR137B, CCL1, ZNFX1, and CDKN1A [30,31]. miR-135a targets JAK2, miR-148a-3p targets HOMER1 and IL-15, and miR-330-3p and miR-450b-5p affect ELF-1 [32,33,34]. The roles of most of these genes in cHL pathogenesis are currently not entirely clear.

In this review, we evaluated the relevance of experimentally validated miRNA–target gene interactions for genes important for the common mechanisms in cHL pathogenesis, i.e., loss of B-cell phenotypes, immune evasion, and growth support. We summarized the published cHL miRNA signatures, retrieved validated miRNA–target gene interactions for selected genes, and evaluated the expression patterns of these miRNA–target gene pairs. Inversed expression patterns were regarded as support for miRNA–target gene interactions that might be relevant for cHL pathogenesis. This overview highlights miRNAs that show a high potential for having critical roles in the pathogenesis of cHL and can be used as a starting point for further studies.

## 2. Methods

We first generated a cHL miRNA signature based on published miRNA profiling studies. All PubMed-indexed papers on miRNAs and Hodgkin lymphoma were screened and from these, we selected all profiling studies. Profiles were generated by small RNA sequencing (RNA-seq) and reverse transcription–quantitative polymerase chain reaction methods (RT-qPCR) on microdissected HRS cells and/or cHL cell lines (Appendix A). Studies using total cHL tissue samples were excluded. We summarized the differentially expressed miRNAs in cHL in comparison to GC B cells, reactive lymph nodes, and non-Hodgkin lymphoma (NHL) cell lines. We also summarized the miRNAs reported as being highly abundant in cHL. In addition, we included EBV-derived miRNAs (EBV-miRNAs) that were reported to be expressed in HRS cells of EBV+ cHL cases.

To identify miRNA–target gene interactions that might be relevant for cHL pathogenesis, we generated a list of genes that are relevant for cHL and termed these “cHL pivotal genes”. These genes were categorized under one of the three themes: loss of B-cell phenotypes (e.g., PAX5, OCT2), immune evasion (e.g., HLA class I, PD-L1), and growth-supporting signals (e.g., NF-κB pathway) (Appendix A).

For the cHL pivotal genes, we conducted a second literature review to identify all miRNAs that have been shown to regulate these genes in HRS cells or any other cell type. We primarily focused on studies involving cancer cells, but also included studies in normal human cells for genes for which cancer-specific references were absent. A reported interaction between a miRNA and the target gene of interest was only accepted when validated by (luciferase) reporter assays or showing changes at the protein level upon miRNA inhibition or overexpression (Appendix A).

Finally, we checked whether the miRNAs that regulate the cHL pivotal genes were present in our cHL miRNA signature and whether they were up- or downregulated. Inverse expression patterns between miRNAs and their respective target genes were considered an indication of a potentially relevant interaction. These miRNA and target gene interactions are further discussed in this review.

## 3. Results

### 3.1. miRNA Signature and Genes Critical for cHL Pathogenesis

Multiple studies have characterized the miRNA expression signature of cHL by studying microdissected HRS cells and cell lines. The focus and results of these studies are summarized in Appendix A [25,34,35,36,37].

Three research groups generated miRNA profiles using cHL cell lines and compared these with GC-B cells [34,36,37], reactive lymph nodes [35], or NHL cell lines [25,34]. Out of 210 miRNAs, 98 were found to be upregulated between cHL cell lines and GC-B cells. Of these, 25 were consistently upregulated in at least two studies and 3 (miR-9-5p, miR-155-5p, and miR-196a-5p) were upregulated in all studies [25,34,36,37]. Conversely, 50 miRNAs were reported to be downregulated in cHL compared with GC-B cells, of which, 8 miRNAs (miR-148-3p, miR-148a-5p, miR-150-3p, miR-150-5p, miR-181a-2-3p, miR-577, miR-598-3p, and miR-3150b-3p) displayed consistent downregulation in at least two of the studies [25,34,36,37]. In one study, isolated primary HRS cells were compared with GC-B cells and this revealed 30 upregulated and 11 downregulated miRNAs [36]. Notably, 12 of these miRNAs displayed consistent upregulation in both HRS cells and cHL cell lines and 3 miRNAs were consistently downregulated [36]. 

Three studies provided a list of highly abundant miRNAs in cHL cell lines. One group pinpointed 26 out of the 183 miRNAs studied as being the most abundant based on RT-qPCR profiling [25]. The other two studies identified a top-10 [37] or top-5 list of the most abundant miRNAs based on small RNA-seq [34]. Notably, miR-142-5p, miR-191-5p, miR-21-5p, miR-92a-5p, miR-155-5p, let-7f-5p, and let-7a-5p were noted as highly abundant in at least two of the studies. 

EBV produces numerous miRNAs that are primarily located in two major clusters in its genome: the BHRF1 (BamHI fragment H rightward open reading frame 1) and the BART (BamHI-A rightward transcript) clusters [38]. The expression of three EBV-derived miRNAs, BART2-5p, BART13-3p, and BART19-3p, has been identified in EBV+ cHL [39,40]. 

Based on these profiling data, we selected a total of 210 human and 3 EBV-derived miRNAs as a starting list for finding miRNA–target gene interactions which are potentially critical for cHL. 

A list of all known validated miRNA–pivotal cHL gene interactions, categorized by the three themes related to cHL pathogenesis (loss of B-cell phenotypes, immune evasion, and growth-supporting signals) is provided in Appendix A. In Table 1 we provide an overview of all miRNA–target genes which based on their expression pattern may be relevant for cHL pathogenesis. These interactions are described in more detail below. 

### 3.2. Loss of B-Cell Phenotypes

The loss of B-cell phenotypes has been linked to the diminished expression of several transcription factors (TFs), including Paired Box 5 (PAX5), octamer-binding transcription factor 2 (OCT2), SPI-1 (PU.1), early B-cell factor 1 (EBF1) and ETS Proto-Oncogene 1 (ETS1) [6,7]. We explored whether deregulated expression of miRNAs could play a role in the aberrant expression of these TFs.

Direct targeting of PU.1 by miR-27a-5p was shown in cHL cell lines by luciferase reporter assays [34]. The overexpression of miR-27a-5p, as observed in cHL cell lines and microdissected HRS cells, fits with a potential role of this miRNA in downregulating PU.1. An interaction between miR-155-5p and PU.1 has been shown in acute myeloid leukemia, in which, the knockdown of miR-155-5p led to increased expression of PU.1 [41]. In HRS cells, miR-155-5p is among the upregulated and highly abundant miRNAs. Thus, the low expression of PU.1 in HRS cells might be induced in part by high levels of miR-27a-5p and miR-155-5p. EBF1 was shown to be targeted by miR-19a-in oral squamous cell carcinoma [42]. As the expression of miR-19a-5p is high in HRS cells, the loss of EBF1 expression in cHL might be caused by miR-19a-5p. Several studies demonstrated a miRNA-dependent regulation of ETS1 expression in a wide variety of diseases and malignancies. The experimentally validated miRNAs potentially relevant for cHL include miR-200c, which was detected in colorectal cancer [43]; miR-9 in gastric carcinoma [44,45]; miR-125b in breast cancer [46]; and miR-155-5p in hematopoietic progenitor cells [47]. The elevated expression of these miRNAs in cHL supports their potential involvement in the decreased levels of ETS1. Currently, no miRNAs with potential relevance for cHL pathogenesis have been described that target PAX5 or OCT2 [48]. 

### 3.3. Immune Evasion

#### 3.3.1. Antigen Presentation

The expression of HLA class I is frequently lost in HRS cells, leading to an impaired presentation of tumor-associated neoantigens. HLA class I loss is more common in EBV-negative cHL patients (~80%) than in EBV-positive cHL patients (~25%) [49]. The loss of HLA class I is mediated by various mechanisms, including the loss of the HLA gene loci and mutations in beta-2 microglobulin (B2M) [50,51,52]. Several miRNAs were reported to regulate the expression of genes associated with antigen presentation, indicating a miRNA-dependent mechanism as an alternative way to reduce the presentation of neoantigens. Three miRNAs were reported to disturb antigen presentation by regulating the peptide transporter proteins TAP1 or TAP2 and the chaperone calreticulin, which all participate in the peptide-loading complex located within the endoplasmic reticulum. miR-346 was demonstrated to directly target TAP1 and thereby reduce HLA class I expression in non-small cell lung cancer cells and cervical cancer cells [53]. miR-125a-5p was shown to regulate TAP2 in esophageal adenocarcinoma cells [54]. miR-27a downregulated HLA class I surface expression through the suppression of calreticulin in colorectal cancer cells [55]. miR-346, miR-125a-5p, and miR-27a are all upregulated in cHL and might therefore contribute to the decrease in antigen presentation in cHL cases without genomic aberrations in the HLA class I and B2M genes. 

The loss of HLA class I makes HRS cells more vulnerable to attack by NK cells. To overcome this, HRS cells upregulate HLA-G [56]. Several studies have pinpointed the miR-148 family (miR-148a, miR-148b, and miR-152) as regulators of HLA-G expression in solid cancers and normal cells [57,58,59,60,61,62]. This miRNA family showed a strong binding affinity to HLA-G transcripts, indicating a potential regulatory effect [60,61,62,63]. Since miR-148a-3p is downregulated in cHL, this miRNA might contribute to the aberrant high expression of HLA-G. Interestingly, miR-16-5p was shown to interact with AGO2 at the HLA-G promoter. This interaction was proposed to result in a non-classical miRNA-mediated activation of the HLA-G gene [64]. This finding suggests that miR-16-5p, which is highly expressed in cHL, may also contribute to the expression of HLA-G. 

The EBV-derived miRNA BART2-5p can affect immune escape by targeting the HLA class I polypeptide-related sequence B (MICB) [65]. MICB is a ligand for the stimulatory natural killer group 2 member D (NKG2D) receptor, which is expressed on NK cells and cytotoxic T cells [66]. MICB is upregulated upon stress exposure such as viral infections, DNA damage, and malignant transformation [67]. The downregulation of MICB by BART2-5p could help EBV+ HRS cells to evade immune recognition.

About 40% of cHL patients show a loss of HLA class II expression in HRS cells, with slightly higher percentages in EBV-negative as compared to EBV-positive cHL patients [12,68]. HLA class II expression has been reported as an independent prognostic marker in cHL [68]. The loss of the HLA class II gene loci and translocation affecting the CIITA locus partly explain the loss of HLA class II expression [69]. One study revealed that miR146b-5p and let-7f-5p both target CIITA in macrophages and in various cancer cell types [70]. The expression of miR-146b-5p is increased in cHL, supporting a potential role of this miRNA in the loss of HLA-II expression. Let-7f-5p is highly abundant in cHL and could also inhibit CIITA expression. 

#### 3.3.2. Immune Checkpoint

The immune checkpoint ligands PD-L1/PD-L2 are highly expressed in a proportion of cHL cases [14]. This has been attributed to copy number gain of 9p24.1, which contains the PD-L1 and PD-L2 gene loci. Co-amplification of the JAK2 gene locus further boosts the expression of PD-L1 by activating JAK/STAT signaling [71]. In addition, the expression of PD-L1 can be induced by the EBV-derived latent membrane protein 1 (LMP1) protein, via activating the JAK/STAT and AP-1 pathways [72]. However, PD-L1 expression is low in most cHL cell lines, with membrane expression being evident by flow cytometry only in SUPHD1 and HDLM-2 cell lines [73,74]. Many studies have highlighted a role for miRNAs in regulating PD-L1 expression across various malignancies. Notably, various miRNAs downregulated in cHL, including miR-20b-5p, miR-34b, miR-138-5p, miR-148a-3p, miR-340-5p, miR-455-5p, and miR-766-5p, were shown to suppress PD-L1 expression in different cancer cell types [75,76,77,78,79,80,81,82,83,84]. The downregulation of these miRNAs in cHL might contribute to the increased expression of PD-L1. Additionally, the miR-200 family, particularly miR-200b and miR-200c, has been identified as key regulators of PD-L1 [85,86]. miR-200b has been implicated in targeting PD-L1 and inhibiting IFN-γ-stimulated PD-L1 expression in gastric cancer, while miR-200c targets PD-L1 in HBV+ human liver cell lines and acute myeloid leukemia [87,88]. As miR-200b is downregulated in cHL, it may play a regulatory role in PD-L1 expression, whereas the upregulated miR-200c is less likely to play a role in cHL. Furthermore, miR-155 exhibits a complex role in the regulation of PD-L1 expression. miR-155 induced a decrease in PD-L1 expression in lymphatic endothelial cells, Hela cells, and dermal fibroblasts [89]. Conversely, it enhanced PD-L1 expression via non-canonical mechanisms by interacting with the 3′UTR of the PD-L1 transcript in diffuse large B-cell lymphoma (DLBCL) [90]. This suggests that the highly expressed miR-155 may also contribute to the high level of PD-L1 in cHL.

#### 3.3.3. Immunosuppressive Molecules

HRS cells overexpress key immunosuppressive molecules like IL-10, TGF-beta, and galectin-1 to create an immune-suppressive microenvironment and prevent effective anti-tumor immune responses [16]. miR-106a has been shown to directly inhibit IL-10 expression levels in Jurkat T cells [91]. miR-106a levels were reported to be downregulated when compared to GC-B and NHL, but miR-106a was also among the highly abundant miRNAs in cHL. It remains to be elucidated whether miR-106a is involved in the upregulation of IL-10 in cHL. miR-130a-3p and miR-143-3p target TGF-beta in HEK-293T cells and mesangial renal cells [92,93]. Both miRNAs are downregulated in cHL, implying that their decreased levels may contribute to the elevated expression of TGF-beta [94,95].

Based on the current knowledge, it is likely that multiple miRNAs play a role in supporting the immune evasion of HRS cells. In Figure 1, we provide an overview of the miRNAs with a potential role in immune evasion mechanisms in cHL.

**Table 1 cancers-16-01126-t001:** Proven miRNA–target gene interactions with potential relevance for cHL pathogenesis.

Related Function	Target Gene	miRNA	Validated Cell Type
Loss of B-cell phenotypes	PU.1 ↓	miR-27a-5p ↑	classical Hodgkin lymphoma [34]
		miR-155-5p * ↑	acute myeloid leukemia [41]
	EBF1 ↓	miR-19a-5p * ↑	oral squamous cell carcinoma [42]
	ETS1 ↓	miR-9 ↑	gastric carcinoma [44]
		miR-125 b ↑	breast cancer [46]
		miR-155-5p * ↑	hematopoietic progenitor cells [47]
		miR-200c ↑	human colorectal cancer [43]
Immune evasion	TAP1 ↓	miR-346 ↑	non-small cell lung cancer, cervical cancer [53]
	TAP2 ↓	miR-125a-5p ↑	esophageal adenocarcinoma [54]
	Calreticulin ↓	miR-27a ↑	colon cancer [55]
	HLA-G	miR-16-5p *^,#^ ↑	breast cancer, colon cancer [64]
		miR-148a-3p ↓	human choriocarcinoma [57], colon cancer [59], oral squamous carcinoma [62];
	CIITA ↓	miR-146b-5p ↑	melanoma, cervical cancer, gastric cancer, macrophages [70]
		let-7f-5p *	melanoma, cervical cancer, gastric cancer, macrophages [70]
	MICB ↓	EBV-BART2-5p	colon cancer [65]
	PD-L1 ↑	miR-20b-5p * ↓	colon cancer, lung cancer, breast cancer [75,76]
		miR-34b ↓	non-small cell lung cancer [77]
		miR-138-5p ↓	colon cancer [78,79]
		miR-148a-3p ↓	colon cancer [80], anaplastic thyroid carcinoma [81]
		miR-155 *^,#^ ↑	HEK-293T cells, (EBV+/EBV−) B-cell lymphoma [90]
		miR-200a ↓	lung cancer [86], breast cancer [85]
		miR-200b ↓	gastric cancer [96], lung cancer [86], breast cancer [85]
		miR-340-5p ↓	cervical cancer [82]
		miR-455-5p ↓	hepatocellular carcinoma [83]
		miR-766-5p ↓	ovarian carcinoma [84]
	IL10 ↑	miR-106a * ↓	T cells [91]
	TGF-β ↑	miR-130a-3p ↓	autoimmune disease [97]
		miR-143-3p ↓	mesangial cells in diabetic nephropathy [93]
Growth-supporting signals	TNF-α ↑	miR-130a-3p ↓	sepsis [98]
	RELA ↑	miR-138 ↓	trophoblasts [99]
		miR-520/373 ↓	breast cancer [100]
	REL (CREL) ↑	miR-181a-5p * ↓	diffuse large B-cell lymphoma [101]
	NFKB1 (P50) ↑	miR-181a-5p * ↓	diffuse large B-cell lymphoma [101]
	NFKBIA (IκBα) ↓	miR-126 ↑	ulcerative colitis [102]
	TNFAIP3 ↓	miR-29c * ↑	hepatocellular carcinoma [103]
		miR-125a/b ↑	diffuse large B-cell lymphoma [104]
		miR-23a-3p ↑	classical Hodgkin lymphoma [34]
	IGF1R ↑	miR-376a ↓	melanoma [105]
		miR-143-3p ↓	nasal squamous cell carcinoma [106], rheumatoid arthritis [107]
		miR-30a-5p ↓	melanoma [108]
	Notch1 ↑	miR-30a ↓	pancreatic beta cells [109], podocytes [110]
		miR-363-3p ↓	gastric cancer [111]
	IL-21R ↑	miR-30a ↓	autoimmune encephalomyelitis [112]

*: abundant miRNA; #: miRNAs shown to promote gene expression, see text for details. ↑/↓: increase/decrease in protein/miRNA expression.

### 3.4. Growth-Supporting Signals

#### 3.4.1. NF-κB Pathway

The activation of NF-κB in cHL may occur via extrinsic signals relayed by cell surface receptors and through genomic aberrations in the components of the NF-κB signaling pathway [6,113,114]. The activation of the NF-κB pathway provides a proliferative advantage to HRS cells via diverse mechanisms, including inhibition of apoptosis and activation of the JAK/STAT pathway [114]. Furthermore, NF-κB signaling can contribute to the observed loss of B-cell phenotypes [114]. In cHL, the triggering of cell surface receptors CD30, CD40, and CD86 induced the activation of the NF-κB pathway. The triggering of CD40 by CD40L expressed on rosetting T cells is one of the main extrinsic factors that activates NF-κB in HRS cells. The NF-κB pathway can also be activated by a ligand, TNF-α, which is secreted by HRS cells into the TME, which results in HRS cell growth [115,116,117,118]. 

There are no reports on miRNAs that regulate CD30, CD40, or CD86 and that have an inversed expression pattern in cHL [119,120]. However, a miRNA-dependent regulation of TNF-α has been shown for miR-130a-3p which is downregulated in cHL, supporting a potential role for this miRNA [98]. Multiple miRNAs have been shown to target different NF-κB subunits in various studies [99,100,101,121]. miR-138 and members of the miR-520/373 family were shown to target RELA in breast cancer [99,100]. The expression of miR-520a and miR-138 are expression in cHL and might facilitate high RELA expression. In another study, miR-181a-5p was shown to exert a direct regulatory effect on REL, NFKBIA (IκBα), and NFKB1, resulting in the repression of NF-κB signaling in DLBCL [101]. miR-181a-5p is downregulated in cHL cells, supporting a direct regulatory role. Another study showed that NFKBIA is targeted by miR-126 in ulcerative colitis [102]. As NFKBIA is an inhibitory component of the NF-κB pathway, the upregulation of miR-126 in cHL profiling studies highlights an important role for this miRNA. Another NF-κB inhibitor, TNFAIP3, is targeted by miR-29c in hepatocellular carcinoma cells [103] and by miR-125a and miR-125b in DLBCL [104]. Interestingly, all miR-29 family members are highly abundant in cHL and both miR-125a and -b are upregulated in cHL. This points towards a potential regulation of TNFAIP3 by these miRNAs in cHL. miR-23a-3p is the only miRNA for which a direct interaction has been shown with TNFAIP3 in cHL cell lines. The inhibitory effect of miR-23a-3p on TNFAIP3 was shown to enhance activation of the NF-κB pathway [34]. Thus, miR-125a/b and miR-23a-3p might facilitate the activation of NF-κB through downregulation of TNFAIP3 in cHL. In addition, miR-21 was reported to the regulate tumor suppressor and regulator of B-cell differentiation BTG2 and E3 ubiquitin ligase PELI1 which are known to regulate c-REL levels [122].

#### 3.4.2. Other Pathways

Receptor tyrosine kinases such as mesenchymal–epithelial transition factor (c-MET) and insulin-like growth factor 1 receptor (IGF1R) are expressed in the majority of cHL cases. They play a pivotal role in promoting cell proliferation and growth by activating the MAPK/ERK and PI3K/AKT pathways. The MAPK pathways involve kinases such as Raf, MEK, and ERK1/2. Upon activation, ERK1/2 translocates from the cytoplasm to the nucleus, and also participates in the phosphorylation of various nuclear transcription factors, such as PU.1 and ETS1 [123,124,125,126,127]. miR-143-3p was shown to downregulate IGF1R in rheumatoid arthritis and in squamous cell carcinoma [106,107]. A direct regulation of IGF1R by miR-30a-5p and miR-376a was shown in melanoma cells [105,108]. The results for miR-30a-5p were inconsistent in profiling studies in cHL with an increased expression in RT-qPCR analyses and a decreased expression in small RNA-seq studies [34,36,37]. This discrepancy highlights the need to investigate the expression of miR-30a-5p in cHL more closely. Regarding IGF1R, a potential regulatory role seems most likely for miR-376a and miR-143-3p in cHL. 

NOTCH1 is highly expressed in cHL, whereas its ligand Jagged1 (JAG1) is present on infiltrating cells. Their interaction supports the proliferation of HRS cells [128]. NOTCH1 is regulated by miR-363-3p in gastric cancer [111]. The decreased expression of miR-363-3p in cHL fits with a potential role in upregulating NOTCH1 levels in HRS cells. 

The stimulation of IL-21R, IL-6R, and CSF1R results in the activation of the JAK/STAT pathway and as such, provides a proliferative advantage to HRS cells [129]. A direct regulation of IL-21R by miR-30a-5p was shown in various studies [109,110,112]. As mentioned above, the results on miR-30a-5p levels in cHL are inconsistent. 

The most likely miRNA–target gene interactions relevant for supporting growth in cHL are shown in Figure 2.

## 4. Discussion

CHL has unique cell survival and disease progression characteristics that are supported by the activation of specific intracellular pathways and interactions between the HRS cells and the TME [4,6,114]. In this review, we highlighted the role of the currently known and potential miRNA–target gene interactions relevant for the pathogenesis of cHL. By comparing the currently available miRNA profiling studies, we generated a list of miRNAs with either an aberrant expression pattern or are highly abundant in cHL. We focused on miRNA target genes that have a role in the main features of cHL, i.e., loss of B-cell phenotypes, immune evasion, and growth-supporting signals. These processes have been linked to cHL pathogenesis in multiple studies. We showed that a miRNA-dependent regulation seems likely for multiple genes based on known experimentally validated targets in any cell type, in combination with inverse expression patterns in cHL. 

Interestingly, several miRNAs may be involved in cHL pathogenesis via targeting multiple genes. miR-155-5p emerges as a central regulator. On the one hand, it can target both PU.1 and ETS1 and thus promote the loss of B-cell phenotypes. On the other hand, miR-155 may enhance the expression of PD-L1 via non-canonical mechanisms. A decrease in miR-148a-3p may facilitate increased levels of HLA-G and PD-L1 and thereby contribute to immune evasion. Likewise, lower levels of miR-130a-3p may contribute to increased levels of tumor necrosis factor-alpha (TNF-α), resulting in growth support and the promotion of an immune-suppressive environment by facilitating expression of TGF-beta. Decreased levels of miR-143-3p may contribute to the elevated expression of TGF-beta and act as a potential regulator of IGF1R. Furthermore, multiple miRNAs that modulate NF-κB signaling also target other genes relevant for cHL. miR-138-5p targets PD-L1 in addition to RELA. miR-125a and miR-125b target, in addition to NF-κB pathway components, TAP2 and ETS1, respectively. Finally, miR-181a-5p has been demonstrated to target multiple NF-κB components. All in all, these miRNAs, with their broad targeting spectrum, underscore the interconnection of the potential molecular mechanisms driving cHL pathogenesis. For 41 of the 49 selected cHL pivotal genes, regulatory miRNAs have been identified in other malignancies or normal cell types. Our review provides detailed descriptions of the potential miRNA regulatory mechanisms for 21 of these 41 genes in cHL. We did not focus extensively on the remaining miRNA targets, because either the miRNAs were not expressed in cHL or they did not exhibit inverse expression patterns.

Our study has some limitations, which may have led to an incomplete overview. First of all, we did not include studies that focused on single candidate miRNAs for the generation of our miRNA profile. The miRNA profiling studies we did use were somewhat limited in overlap. Discrepancies in differential expression might have been caused by differences in the starting material (cHL cell line/microdissected HRS cells), control samples (GC-B/NHL/RLN cells), and methodology (small RNA-seq/RT-qPCR). In addition, it is possible that although specific miRNA–target gene interactions were validated in a certain cell type, they may not be relevant in HRS cells. Differences between effects in different cell types can be caused by the unavailability of miRNA binding sites or differences in abundance of other targets of the miRNA. Finally, we limited our target gene list to 49 genes that we considered pivotal within the three selected main themes, potentially leaving out genes that may also be important for cHL pathogenesis. 

In our review, we highlighted validated miRNA–target gene interactions with potential relevance for cHL based on inverse expression patterns of the miRNAs and the selected target genes. This provides a starting point for future studies aiming to define the role of miRNAs in cHL. These studies should experimentally validate the proposed miRNA–target interactions in cHL and explore the downstream effects. The appropriate wet lab approaches include luciferase reporter assays and effects on target gene protein levels upon modulation of miRNA expression or mutations in the endogenous miRNA binding sites in cHL cell lines. In addition to single miRNA approaches, it may also be interesting to consider genome-wide approaches using miRNA gain- or loss-of-function screens. As these approaches are unbiased, they have the advantage of not only validating known interactions identified in other cell types, but could also discover new interactions. By identifying all miRNAs that can change the expression of a specific marker or activity of a specific pathway, comprehensive lists of miRNAs relevant for the phenotype can be readily identified. Recent studies suggest that the application of miRNA overexpression and CRISPR knockout screens offers a reliable and feasible approach to perform such genome-wide screens [130,131]. An important next step will be to study the expression of miRNAs and their target proteins in primary tissues. This can be accomplished by combining miRNA in situ hybridization with immunohistochemistry to show a potential inverse correlation.

An interesting follow-up question is whether this knowledge can be used as a starting point for the development of novel therapeutic approaches. A variety of clinical trials have been tested for miRNA therapeutics and efficient delivery methods in recent years. These methods include delivery routes via liposomes, polymers, extracellular vesicles, or nanoparticles. In addition to the delivery methods, the functional implications, such as the kinetics, dosing, and the targeting of the miRNAs have been tested in preclinical and clinical trials [132]. 

In conclusion, this review underscores the intricate involvement of multiple miRNAs in cHL pathogenesis. Notably, miR-155-5p, miR-148a-3p, miR-130a-3p, miR-143-3p, miR-138, miR-125a, miR-125b, miR-181a-5p, and miR-23a-3p influence crucial aspects of cHL pathogenesis including loss of B-cell phenotypes, immune evasion, and growth-supporting pathways by targeting various genes. Overall, these miRNAs provide a foundation for future studies aiming to further explore the relevance of miRNAs in cHL pathogenesis. Experimental validation and potential translation into therapeutic strategies may open new avenues for targeted interventions in cHL.

## Figures and Tables

**Figure 1 cancers-16-01126-f001:**
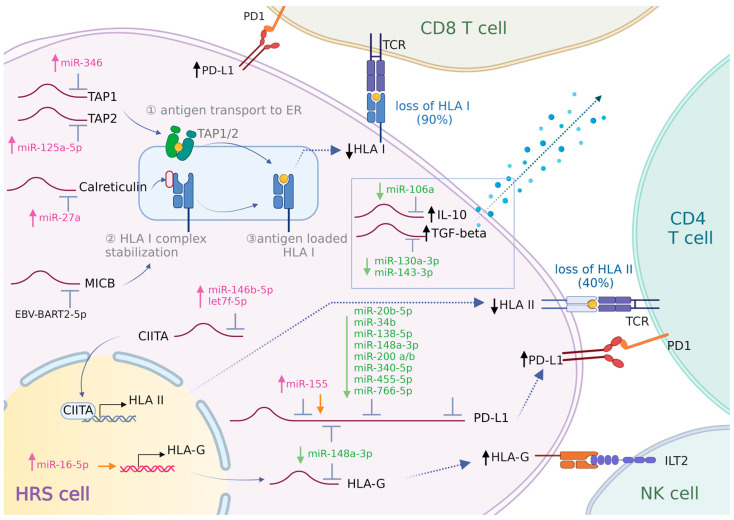
Involvement of miRNAs in HRS cell immune evasion. HRS cells employ various strategies to escape immune surveillance. (1) HRS cells achieve a reduction or loss of neo-antigen presentation by downregulating HLA-I and/or HLA-II. The process of loading HLA-I onto the cell membrane involves several key proteins. TAP1 and TAP2 transport peptides into the endoplasmic reticulum (ER). Inside the ER, calreticulin and B2M assist in the proper loading of peptides onto HLA-I molecules. Once fully assembled and loaded, these HLA-I molecules are then transported to the cell surface. CIITA is a transcriptional activator of HLA-II. Deregulation of the expression of these key proteins could impair tumor antigen presentation and T-cell activation. (2) HRS cells upregulate HLA-G to counterattack NK cell activation. (3) HRS cells highly express PD-L1 to enhance PD-L1/PD-1 signaling, thereby inducing T-cell “exhaustion”. (4) HRS cells create an immunosuppressive microenvironment and inhibit the activation of cytotoxic T cells by secreting galectin-1, IL-10, and TGF-beta. The role of miRNAs in modulating the expression of these proteins is shown. ↑/↓(black): increase/decrease in protein expression. miRNAs shown in pink/green: upregulated/downregulated expression. Orange arrow: miRNA promoting gene expression.

**Figure 2 cancers-16-01126-f002:**
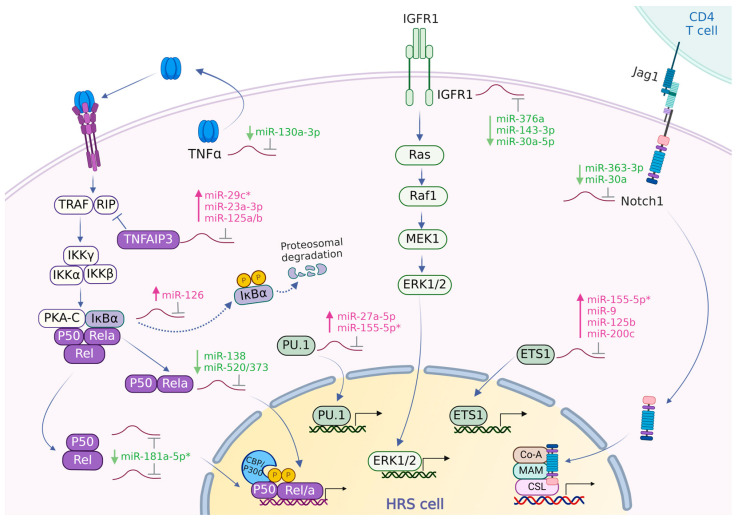
miRNA–target gene interactions supporting growth of HRS cells. Schematic representation of the NF-κB, MAPK/ERK, and Notch signaling pathways, along with the potentially relevant regulatory roles of miRNAs. NF-κB activation is facilitated by, among other factors, tumor necrosis factor-alpha (TNF-α). This activation involves the phosphorylation and proteasomal degradation of IκBα, leading to the nuclear translocation of RelA and other Rel proteins. TNF-α, secreted by HRS cells into the tumor microenvironment (TME), can activate the NF-κB pathway in an autocrine manner. IGF-1R activation contributes to the phosphorylation and nuclear translocation of ERK1/2, which are crucial for regulating cell growth. Additionally, the expression of the transcription factors PU.1 and ETS1 is also regulated by ERK1/2. Notably, Notch signaling is activated by the nuclear translocation of Notch1. Overall, the activation of the NF-κB, MAPK/ERK, and Notch signaling pathways collectively enhances the survival and proliferation of HRS cells. miRNAs shown in pink/green text: upregulated/downregulated miRNAs. *: abundant miRNA.

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
