# Peer review of "Pinpointing Functionally Relevant miRNAs in Classical Hodgkin Lymphoma Pathogenesis"

_cancers, 2024, doi:10.3390/cancers16061126_

Round 1

Reviewer 1 Report

Comments and Suggestions for Authors

Pan and colleagues provide in their review a concise overview of deregulated miRNA expression in Hodgkin lymphoma and discuss the protein-coding genes likely regulated by these miRNAs. They group the genes in three functional categories, namely lost B cell phenotype, immune evasion and growth supporting signals. 

This is a valuable overview.

Minor points

1. Line 12:"our review organized published human miRNA profile" is not good English. Rephrase. In addition, add "are" before "crucial" in this line.

2. Line 15: it should read: "genes, our review" and "enhances the".

3. Lines 18-19: "a malignancy... called Hodgkin and Reed-Sternberg (HRS) cells" is a wrong phrasing.  

4. Line 39: The WHO classification from 2017 cited as ref. 2 is somewhat outdated. A discussion of the new 5th WHO lymphoma classification has already been published in Leukemia last year.

5. Lines 44-45: PAX5 expression is not lost in HRS cells, but only downregulated. Weak protein expression can typically be detected by standard immunohistochemistry.

6. Line 46: Although only few normal B cells express CD30, CD30 is nevertheless also physiologically expressed at distinct B cell differentiation stages (see Weniger et al., J Clin Invest 2018).

7. Line 53: about half of HL cases also express PD-L2, which might be worthwhile mentioning here.

8. Line 58: It is recommended to write throughout the text NF-kB with "k" as greek symbol for kappa. This is the typical writing for this.

9. Line 60: two commas after IL-9.

10. Line 62: double empty space after "cells". There are several additional instances of this (e.g., lines 297, 304).

11. Lines 65-66: It is here only stated that miRNAs inhibit translation. It should be specified that miRNA often also cause degradation of their target mRNAs.

12. Line 66: It should be specified that the target molecules are mRNAs.

13. Line 86: "important for" better than "important to"?

14. Line 100: "Tables" instead of "Table".

15. Line 107: insert "of" between "one" and "the".

16. Line 254: "miRNAs" missing after "abundant".

17. Figure 1: the word "miR-106a" near IL-10 should be written in green, as it is downregulated according to the arrow.

18. Line 279: It seems that the notes to the Table are incomplete. Moreover, I did not detect any underlined miRNA. In addition, "by directly binding" of what?

19. Line 280: correct "signalis".

20. Line 322: delete"." after ETS1.

21. Line 35: I see only single arrow besides a protein (Ras). Have other arrows been forgotten?

22. Line 455: author list strange, double names and multiple "Fau".

23. Missing page ranges or article numbers for several references: 62 69, 86, 95,118. Please check all references.

24. Ref 123: Ku and Bra correct names?

Reviewer 2 Report

Comments and Suggestions for Authors

In the current manuscript, the authors comprehensively summarized the expression of microRNAs (miRNAs) in classical Hodgkin lymphoma. (cHL). The manuscript is well summarized and well written. As the authors described in Discussion section, I agree that one of the limitations in theses studies is the differences in starting materials for the analysis. This point has been clearly described in the manuscript.

Author Response

We would like to thank the reviewers for their positive comments.

Reviewer 3 Report

Comments and Suggestions for Authors

The present manuscript provides an overview of miRNAs directly involved in the pathogenesis of classical Hodgkin lymphoma by regulating many important genes or signalling pathways. In general, the manuscript is well-written, detailed, clear and well-organized. I found no inaccuracies or significant omissions of literature. I think this review can be published in its current form.

Author Response

(The authors gave the same response as above.)
